# Antecedents of Workplace Bullying among Employees in Germany: Five-Year Lagged Effects of Job Demands and Job Resources

**DOI:** 10.3390/ijerph182010805

**Published:** 2021-10-14

**Authors:** Paul Maurice Conway, Hermann Burr, Uwe Rose, Thomas Clausen, Cristian Balducci

**Affiliations:** 1Department of Psychology, University of Copenhagen, 1165 Copenhagen, Denmark; 2Federal Institute for Occupational Safety and Health (BAuA), 10317 Berlin, Germany; burr.hermann@baua.bund.de (H.B.); rose.uwe@baua.bund.de (U.R.); 3National Research Centre for the Working Environment, 2100 Copenhagen, Denmark; tcl@nfa.dk; 4Department of Psychology, University of Bologna, 40127 Bologna, Italy; cristian.balducci3@unibo.it

**Keywords:** workplace bullying, S-MGA, job demands-resources model, psychosocial working conditions, prospective study

## Abstract

Objectives: The aim of the present study was to examine the long-term association of job demands and job resources with self-reported exposure to workplace bullying in a representative sample of employees in Germany. Methods: We analysed a nation-wide representative cohort of employees working in the same workplace with a 5-year follow-up (S-MGA; *N* = 1637). The study contained self-reported measures of psychosocial working conditions, including work pace, amount of work, influence at work, role clarity and quality of leadership, and workplace bullying, and of organisational factors, including organisational restructuring and layoffs. Results: After controlling for bullying and occupational level at baseline, higher baseline levels of organisational restructuring (OR 1.73; 95% CI 1.10–2.70), work pace (1.30; 95% CI 1.01–1.66), and amount of work (1.55; 95% CI 1.21–1.99), and lower baseline levels of influence at work (0.70; 95% CI 0.55–0.90) and quality of leadership (0.64; 95% CI 0.50–0.82), were associated with an elevated risk of workplace bullying at follow-up. In all, 90% of cases of self-reported workplace bullying could be attributed to these factors. Conclusions: The study suggests that employees reporting higher demands and lower resources, as well as organisational factors such as restructuring, are at a higher long-term risk of being targets of workplace bullying. Interventions aimed at preventing workplace bullying could benefit from a focus on psychosocial working conditions and organisational factors.

## 1. Introduction

An increasing number of prospective studies has pointed to a causal link between workplace bullying (also referred to as mobbing) and reduced mental health, especially depression [1,2,3,4,5]. Workplace bullying is also associated with costs for workplaces and societies, as it increases the risk of turnover [6,7], sickness absence [8,9], and disability retirement [10,11]. The identification of possible antecedents of workplace bullying is therefore crucial to inform preventive actions aimed at countering the phenomenon and its adverse impact on mental health. As a contribution to this, we set out to examine whether baseline psychosocial working conditions operate as risk factors for the long-term occurrence of workplace bullying.

According to a widely accepted definition, workplace bullying means “harassing, offending, socially excluding someone or negatively affecting someone’s work. In order for the label bullying (or mobbing) to be applied to a particular activity, interaction or process it has to occur repeatedly and regularly (e.g., weekly) and over a period of time (e.g., about six months)” [12]. Central definitional elements of workplace bullying are therefore (a) an employee’s persistent and repeated exposure to inappropriate treatment by one or more co-workers and/or managers, (b) the feeling of being a victim and (c) the feeling of not being able to defend oneself as a result of the imbalance of power that typically occurs between the target and the perpetrator [13,14].

To date, a substantial body of research has been produced that focuses on potential work-related antecedents of workplace bullying, including job design, management practices, and social context [15,16,17]. This approach, known as the work environment hypothesis [18], conceives workplace bullying as a behavioural stress response stemming from an employee’s experience of a poor psychosocial work environment [19]. Overall, the work environment hypothesis has enjoyed a substantial amount of empirical support in the literature [15,16].

Over the last two decades, the job demands-resources model [20] has been a dominant theoretical approach in empirical investigations of the relationship between a poor psychosocial work environment and employees’ stress responses. This JD-R model posits that high job demands are mainly responsible for generating stress responses (e.g., burnout); in opposition to this, job resources are conceived primarily as protective factors, since they may both reduce the strain generated by high demands and assist employees in achieving their work goals. The mere availability of job resources is also expected to promote work engagement, by stimulating personal growth, learning and development [21]. The theoretical premises of the JD-R model have been substantially supported in empirical research, including longitudinal studies [22].

In the domain of workplace bullying, previous studies employing the JD-R model have supported the work environment hypothesis by stressing the role of job demands and job resources as risk and protective factors, respectively [23,24,25,26,27,28,29,30]. The mechanisms behind these are diverse [19]. On the one hand, the strain resulting from having to deal with excessive job demands may drain employees’ energy reservoirs, impinging on their coping resources and making them “easy targets” of aggressive acts that co-workers may engage in as a way to vent their frustrations [23]. In their effort to deal with high job demands, employees may also become more prone to adopt behaviours that violate workplace norms and expectations, and in so doing trigger negative reactions from co-workers [31]. On the other hand, the availability of job resources (e.g., autonomy in organising one’s working time and a supporting social milieu) may equip employees with more opportunities to experience and actively build positive social interactions at work and/or to prevent problematic encounters from evolving into bullying [31].

Previous studies focusing on work-related antecedents of bullying present mainly three limitations. First, most existing studies are based on cross-sectional designs [25,26,27,29,30], providing limited evidence about cause–effect relationships [32,33]. In particular, the direction of the associations observed between adverse psychosocial working conditions and workplace bullying cannot be established in cross-sectional studies, thereby preventing one from ruling out reversed effects. Second, in a number of studies (e.g., [25]), different demands and resources were combined to form higher order scales, which limits the identification of specific antecedents—a crucial piece of information when it comes to improving psychosocial working conditions potentially leading to workplace bullying. A third and final limitation is that previous longitudinal research focusing on the link between psychosocial working conditions and workplace bullying has mainly adopted short to medium follow-up intervals, such as six months [23], one year [24,34,35], or two years [36]. In accordance with the work environment hypothesis, workplace bullying is an escalating phenomenon resulting from the strain that may develop among employees experiencing adverse psychosocial working conditions [18,37]. However, poor working conditions might take time to produce strain-related consequences [38], which, in turn, might require additional time to affect an employee’s job situation to the point that the process leading to workplace bullying is triggered (e.g., through the above-mentioned easy target or violation mechanisms). Thus, longer time intervals might be useful to reflect the temporal window needed for poor working conditions to exert their expected effect on workplace bullying.

### The Present Study

The aim of the present study was to apply the JD-R model to workplace bullying in a prospective study based on a representative sample of the German working population. Workplace bullying is a prevalent risk factor in Germany, with around 3% of all employees reporting exposure to severe bullying (i.e., at least weekly over a ≥ 6 months period) [39]. In the German context, workplace bullying is seen as an organizational issue that needs to be tackled from an occupational health perspective [40].

We set out to investigate the role of self-reported measures of key distinct work characteristics, which were previously found to be important antecedents of exposure to workplace bullying [16]. Specifically, we examined the associations of baseline psychosocial working conditions, including job demands (i.e., work pace and amount of work) and job resources (i.e., influence at work, role clarity and quality of leadership), and baseline organisational factors, including organisational restructuring and layoffs, with self-reported exposure to workplace bullying measured five years later. Analyses were adjusted for self-reported exposure to workplace bullying and other potential confounders at baseline. To reduce potential biases in risk estimation, we limited the analyses to respondents who remained in the same workplace during the five-year follow-up period.

## 2. Materials and Methods

### 2.1. Sample

We employed the Study on Mental Health at Work (S-MGA), a German nation-wide representative cohort. At baseline (years 2011/2012), the target population was defined as all employees in Germany born in 1951–1980 in employments, for which they had to pay social security contributions as of 31 December 2010; therefore, civil servants, self-employed individuals, and freelancers and people in precarious part-time work were not covered in the study. A total of 13,590 people were randomly selected [41]. All data was gathered based on personal computer-assisted interviews performed in the respondents’ home [41]. Of the 4511 participants who participated in the interviews at baseline (response rate: 33%), 4203 were employed at baseline (on average 1 year after drawing the target population), of which 2485 people participated at follow-up (year 2017; response rate: 59%). Cohort participation did not differ by gender. Younger participants took part to a lower degree than their older counterparts (31–36 years: 15% vs. 49–60 years: 20%); participation among unskilled and skilled workers was lower (4% and 17%, respectively) than participation among semi-professionals, professionals and managers (24%) [41,42].

Included in the present study were only participants who were employed at baseline, did not change employment between baseline and follow-up and did not present missing values at baseline for the main study variables. Assuming a higher stability of working conditions among participants who remained in the same workplace, we excluded those employees who changed employment during the study to reduce the risk of obtaining biased estimates due to changes in exposure occurring between the two measurement points (an approach that was used previously [36]). The application of these exclusion criteria led to a final sample of 1738 participants. Figure 1 shows a flow diagram of participation.

### 2.2. Measures

Self-reported exposure to workplace bullying at baseline and follow-up was assessed using the following two questions: (1) “Do you frequently feel unjustly criticized, hassled or shown up in front of others by co-workers?” and (3) “Do you frequently feel unjustly criticized, hassled or shown up in front of others by superiors?”, with the response options “yes” and “no”. Each question was followed by the question: (2,4) “And how often did it occur in the last 6 months?” with the following response options: “daily”, “at least once a week”, “at least once a month” and “less than once a month”. The predictive validity of this hybrid approach, which combines the behavioural experience and self-labelling methods [39], corresponded to that demonstrated by the reporting of negative acts based on the behavioural experience method alone [43]. Following the aforementioned widely accepted definition of workplace bullying [12], we created a dichotomous workplace bullying variable, where participants were considered as exposed to bullying if they have reported being bullied by co-workers or superiors, or both, “daily” or “at least once a week”. A similar dichotomization was employed in previous studies (e.g., [17]). Moreover, it has been shown that severe forms of exposure, as represented by frequent bullying, are particularly detrimental to targets’ mental health [4].

Working conditions at baseline were assessed by two items measuring organisational restructuring and organisational layoffs—both taken from the German Employee Survey [44]—four scales measuring amount of work, influence at work, role clarity and quality of leadership, and one item measuring work pace—all taken from the Copenhagen Psychosocial Questionnaire [45,46]. Details for each item and the scale included are provided below:

Organisational restructuring: “Was there any fundamental restructuring or reorganization in your immediate work environment?”.

Organisational layoffs: “Were there any dismissals in your immediate work environment within the last 2 years?”.

Both items measuring organisational restructuring and organisational layoffs had a dichotomous response option ‘yes’ or ‘no’.

Amount of work: “How often …-is your workload unevenly distributed so it piles up?”, “-do you not have time to complete all your work tasks?”, “-do you get behind with your work?”, “-do you have enough time to complete all your work tasks?” (4 items, Cronbach’s alpha = 0.81, range of inter-item correlations: 0.23–0.69).

Work pace: “How often do you have to work very fast?”.

Influence at work: “How often …-do you have a large degree of influence on the decisions concerning your work?”, “-do you have a say in choosing who you work with?”, “-can you influence the amount of work assigned to you?”, “-do you have any influence on what you do at work?” (4 items, Cronbach’s alpha = 0.69, range of inter-item correlations: 0.30–0.42).

Role clarity: “Do you know exactly how much say you have at work?”, “Does your work have clear objectives?”, “Do you know exactly which areas are your responsibility?” (3 items, Cronbach’s alpha = 0.70, range of inter-item correlations: 0.37–0.53).

Quality of leadership: “To what extent would you say that your immediate superior …-makes sure that the individual member of staff has good development opportunities?”, “-gives high priority to job satisfaction?”, “-is good at work planning?”, “-is good at solving conflicts?” (4 items, Cronbach’s alpha = 0.85, range of inter-item correlations: 0.54–0.66).

The following five response options were used for items measuring work pace, amount of work and influence at work had: “always”, “often”, “sometimes”, “seldom”, “never/hardly ever”. The items measuring role clarity and quality of leadership had the following five response options: “to a very large extent”, “to a large extent”, “somewhat”, “to a small extent”, “to a very small extent”. For the scales, we calculated the scores as the mean of the component items.

Baseline covariates were gender, age and occupational level. The latter was categorized, based on the International Standard Classification of Occupations (ISCO 08), into the following four groups according to skill level: unskilled workers, skilled workers, semi-professionals, academics/managers [47] (managers were grouped together with academics as in similar classifications [48]).

To check if participants were in the same job between the two measurement points, we asked them the following question at follow-up: “Are you still working in the same position [as stated in the last interview at baseline]?”.

### 2.3. Statistical Analysis

The associations between working conditions at baseline and workplace bullying at follow-up were estimated by means of multiple logistic regression models, calculating odds ratios and their 95% confidence intervals. The variables were entered in successive steps. In the first step (Model 1), each working condition was entered separately, without adjusting for any covariate; in the second step, each working condition was adjusted for occupational level (Model 2); in the last step (Model 3), each working condition was adjusted for both occupational level and workplace bullying at baseline. We entered only occupational level and workplace bullying as baseline covariates based on the analyses shown in Appendix A (Table A1 and Table A2).

We estimated the risk of workplace bullying at follow-up attributable to the exposure to those working conditions significantly associated with workplace bullying in model 3 of the aforementioned logistic regression analysis. For each participant, the mean across the significant working conditions was then calculated to compute an “adverse work environment” index ranging from 0 to 4, with higher scores indicating a more adverse psychosocial work environment. We then created three categories based on the index scores: “Low” (0 to <1.5); “Medium” (1.5 to <2.5) and “High” (2.5 to 4). Participants in the last two categories (scores from 1.5 to 4) were considered exposed to an adverse work environment. Using multiple logistic regression, we calculated odds ratios and their 95% confidence intervals for the association between the adverse work environment index categories at baseline and workplace bullying at follow-up, adjusting for baseline workplace bullying and occupational level. The attributable fraction of workplace bullying due to exposure to an adverse work environment was computed based on the obtained odds ratios and the prevalence of the three categories of the “adverse work environment” index, according to the method developed by Miettinen [49] and Rothman [50].

To examine the role of working conditions in predicting new cases of workplace bullying, we ran the same logistic regression analyses, but in a sub-sample wherein those participants bullied at baseline were excluded (*N* = 1639).

All analyses were performed by means of SPSS 27, using the LOGISTIC REGRESSION command.

## 3. Results

The characteristics of the study sample are shown in Table 1. The sample was equally distributed between women and men. Employees aged 41–50 years and skilled workers represented 45 and 40 percent of the sample, respectively. Around half of the employees experienced organisational restructuring, while two thirds experienced layoffs. On average, the highest scale score was obtained for role clarity (mean 3.33, in a scale ranging from 0 to 4) and the lowest for amount of work (mean 1.71). In all, 6% of the sample reported exposure to workplace bullying at both baseline and follow-up.

Person-r intercorrelations between working conditions at baseline are shown in in Appendix A Table A2. The results of the associations between working conditions at baseline and self-reported exposure to workplace bullying at follow-up are shown in Table 2. In the crude model (Model 1), reporting organisational restructuring, and higher levels of amount of work and work pace at baseline, were associated with a significantly higher risk of workplace bullying at follow-up. On the contrary, higher levels of quality of leadership, influence at work and role clarity at baseline were associated with a significantly lower risk of workplace bullying. Organisational lay offs was the only factor not significantly associated with workplace bullying. The associations remained similar when adjusting for occupational level as baseline (Model 2). When adjusting for workplace bullying at baseline (Model 2b), the associations slightly decreased, and in the final model (Model 3), wherein we additionally adjusted for workplace bullying at baseline, we obtained similar results as in Model 2, although the association between role clarity and workplace bullying became non-significant.

Table 3 shows the associations between the “adverse work environment” index at baseline and workplace bullying at follow-up. The following working conditions were used to compute the index, as described in the statistical analysis section: organizational restructuring (the response option “yes” was assigned the value 3, “no” was assigned value 1), work pace, amount of work, influence at work (reversed score) and quality for leadership (reversed score). Overall, the “adverse work environment” index at baseline was significantly associated with workplace bullying at follow-up (*p* = 0.000). The estimated attributable fraction of workplace bullying at follow-up due to the exposure to an adverse work environment (index score of at least 1.5) was 90%. Under the assumption that a causal link exists, this value indicates that 90% of the bullying cases could be prevented if the participants moved from the two adverse work environment categories (Medium and High) to the least adverse one (Low).

In the analysis wherein only participants not experiencing workplace bullying at baseline were included (Table 4), the associations between baseline working conditions and workplace bullying at follow-up were very close to the associations observed in the main analysis. However, work pace was no longer significantly associated with workplace bullying.

## 4. Discussion

The present study suggests that exposure to job demands, in the form of organisational restructuring, work pace and amount of work, is related to an increased risk of being exposed to workplace bullying five years later. In contrast to this, job resources, including quality of leadership and influence at work, acted as protective factors against exposure to workplace bullying. The exposure to these working conditions accounted for a substantial proportion of the self-reported bullying observed in the present study.

Despite several methodological differences, our study aligns with findings of a few previous longitudinal studies that adopted shorter time intervals. For instance, a six-month follow-up study [23] found a lagged effect of high workload and low job autonomy on being a target of bullying. Similar findings were previously reported in other studies adopting a one-year time interval [24,34,35] as well as in a three-year follow-up study [28]. In another study using a two-year time interval [36], however, the authors found support only for reversed causation (i.e., an effect of workplace bullying on later exposure to poor working conditions). Overall, the present study provides an additional piece of evidence in support of the work environment hypothesis of workplace bullying [18], which conceives the latter as the product of poor psychosocial working conditions. Adding to the available evidence, however, our study adopted a longer time interval that is likely to fit more closely with how the bullying process unfolds over time [18,37]. Indeed, a long time interval might be necessary for poor psychosocial working conditions to produce strain-related effects; these, in turn, might require a certain amount of time to trigger those work-related conditions leading up to the initiation and development of the workplace bullying process [38].

### Methodological Considerations

The use of a prospective design and the inclusion of a range of occupations and industries in a representative sample of the German working population are major strengths of this study.

This study also presents a number of limitations. One main one relates to the five-year follow-up we adopted to examine the impact of psychosocial working conditions on workplace bullying. As systematic methodological considerations of optimal follow-up intervals in studies focusing on the link between psychosocial working conditions and workplace bullying are lacking, it is unclear whether the use of shorter or longer intervals would lead to different estimations of the impact of working conditions on workplace bullying. With regard to the five-year follow up we employed in the present study, other unaccounted for mechanisms might have occurred during this time interval, affecting the exposure or the outcome, and thus biasing the associations observed in our study. For instance, the exposure to bullying behaviours might have declined or disappeared during follow up, which would lead to an effect underestimation. By excluding those employees who remained in the same job from baseline to follow-up, we have limited the chance of obtaining biased estimates due to changes in exposure to working conditions over time, although variations in exposure cannot be completely ruled out. However, in the cohort employed in the present study, the same psychosocial working conditions were correlated more strongly from baseline to follow-up among those who maintained than among those who changed job [51].

Other limitations are related to sample composition, sample size and mode of data collection. First, employees younger than 31 years and employees who were not subjected to social security contribution (civil servants, self-employed individuals, freelancers and people in precarious part time work) were not included in the sample. In particular, the group in precarious part-time jobs (defined as mini- and midi-jobbers in German) is large and would mostly include unskilled and skilled employees who, according to the present study, seem to be at an elevated risk of workplace bullying (see Appendix A). Second, workplace bullying is a low base-rate phenomenon, with a prevalence rate of 5% in our study, which might limit statistical power. Third, we assessed both exposures and outcomes using self-reports obtained in a personal interview setting. To some extent, we considered this by adjusting for baseline levels of workplace bullying, which might have reduced the impact of experienced negative acts on the self-reporting of working conditions. We also performed sensitivity analyses wherein we excluded employees reporting workplace bullying at baseline, and found the same associations as in the main analysis.

Another limitation is that we did not account for personality characteristics of targets, which might influence self-reported exposure to workplace bullying [25]. Finally, as is common in this line of research, we measured working conditions at the individual level. Nevertheless, employing a multi-level approach with analyses at the group level might prove valuable, as employees belonging to the same unit tend to share similar working conditions. In this way, it would be possible to estimate the impact of shared working conditions on the individual risk of workplace bullying [15]. A group-level analysis would also reduce biases in the reporting of working conditions due to individual factors.

## 5. Conclusions

We prospectively examined psychosocial and organisational working conditions as potential antecedents of workplace bullying in a sample of employees in Germany. Instead of employing higher-order scales of job demands and job resources combining different factors together, we examined different working conditions separately, which allowed us to pinpoint specific work-related factors that could increase–or reduce–the risk of workplace bullying. This approach provides more precise indications about the factors that need to be addressed in interventions targeting workplace bullying. Specifically, we found that organisational restructuring, higher work pace and higher amount of work are associated with an elevated risk of workplace bullying. On the contrary, the risk of workplace bullying is decreased when employees experienced higher levels of influence at work and quality of leadership. These findings, which strengthen the conclusions of previous studies [15], suggest that increased efforts to improve the psychosocial work environment are needed in contemporary workplaces as a crucial means for reducing the incidence of mental health outcomes—especially depression—which are established outcomes of exposure to workplace bullying [1,2,3,5,9,52]. This is imperative in light of previous studies showing the detrimental consequences of depression for employees’ workability [53], companies [54], and the social security system [55]. Workplace initiatives to reduce workplace bullying are thus likely to provide benefits at multiple levels, including individuals, workplaces and societies.

## Figures and Tables

**Figure 1 ijerph-18-10805-f001:**
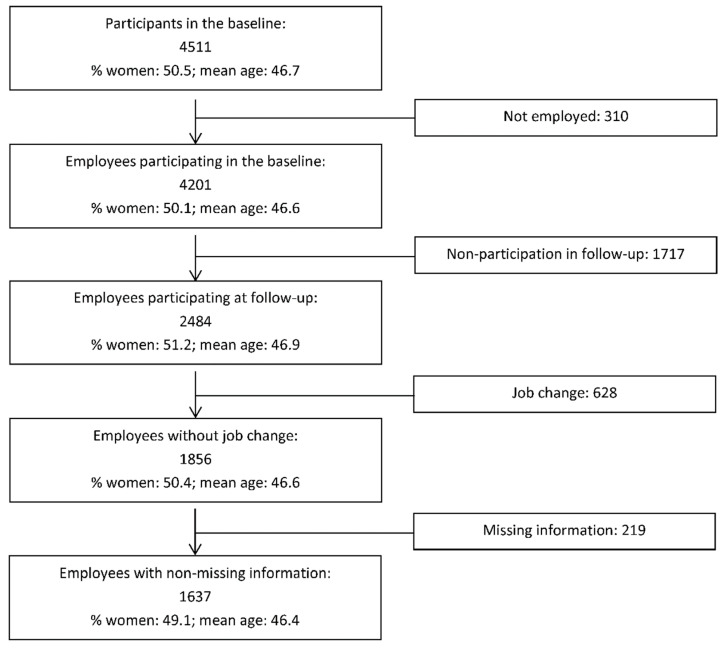
Flow diagram of participation.

**Table 1 ijerph-18-10805-t001:** Characteristics of the sample at baseline (*N* = 1738).

	*N*	%	Mean	SD	Cronbach’s Alpha	Inter-Item Correlations
Gender						
Men	870	50				
Women	868	50				
Age						
31–40 years	388	22				
41–50 years	777	45				
51–60 years	573	33				
Occupational level						
Unskilled workers	101	6				
Skilled workers	692	40				
Semi-professionals	516	30				
Academics/managers	429	25				
Organisational restructuring						
Yes	822	47				
No	916	53				
Organisational layoffs						
Yes	571	33				
No	1167	37				
Work pace *			2.62	0.96	†	†
Amount of work *			1.78	0.92	0.84	0.46; 0.69
Influence at work *			1.73	0.94	0.69	0.30; 0.42
Role clarity *			3.31	0.57	0.71	0.38; 0.53
Quality of leadership *			2.31	0.89	0.84	0.52; 0.64
Bullying						
Yes	99	6				
No	1639	94				
FOLLOW-UP MEASURES						
Bullying						
Yes	95	5				
No	1643	95				
TOTAL	1738	100				

* Scale scores ranging from 0 to 4. † N/A. Measure based on one item.

**Table 2 ijerph-18-10805-t002:** Associations between working conditions at baseline and workplace bullying at follow-up (*N* = 1738).

	Model 1 ^a^			Model 2a ^b^			Model 3 ^c^		
	* **p** *	**OR**	**95% CI**	* **p** *	**OR**	**95% CI**	* **p** *	**OR**	**95% CI**
Organisational restructuring *	**0.020**	**1.64**	**1.08**; **2.50**	**0.010**	**1.76**	**1.15**; **2.69**	**0.017**	**1.73**	**1.10**; **2.70**
Organisational layoffs *	0.050	1.52	1.00; 2.31	0.066	1.49	0.97; 2.27	0.246	1.30	0.83; 2.03
Work pace †	**0.000**	**1.54**	**1.21**; **1.97**	**0.001**	**1.50**	**1.18**; **1.91**	**0.039**	**1.30**	**1.01**; **1.66**
Amount of work †	**0.000**	**1.63**	**1.30**; **2.06**	**0.000**	**1.83**	**1.44**; **2.32**	**0.001**	**1.55**	**1.21**; **1.99**
Influence at work †	**0.000**	**0.59**	**0.46**; **0.75**	**0.000**	**0.64**	**0.51**; **0.82**	**0.006**	**0.70**	**0.55**; **0.90**
Role clarity †	**0.046**	**0.70**	**0.40**; **1.00**	**0.048**	0.70	0.50; 1.00	0.317	0.84	0.59; 1.19
Quality of leadership †	**0.000**	**0.50**	**0.40**; **0.62**	**0.000**	**0.50**	**0.40**; **0.63**	**0.000**	**0.64**	**0.50**; **0.82**

Bold numbers indicate significance level < 0.05. * Dichotomous measure: yes vs. no. † Scale: OR for each one-step increase in a scale score ranging from 0 to 4. ^a^ Each working condition unadjusted. ^b^ Additionally adjusted for occupational level only. ^c^ Additionally adjusted for baseline workplace bullying.

**Table 3 ijerph-18-10805-t003:** Associations between the “adverse work environment” index at baseline and workplace bullying at follow-up (*N* = 1738).

	N (Fraction of Total, %)	Cases of Workplace Bullying at Follow-Up %(n)	*p* ^a^	OR ^b^	95% CI
Adverse work environment index ^a^			**0.000**		
Low (0–<1.5)	226	0.4		1	
Medium (1.5–2.5)	1144	4.4		**8.69**	**1.19**; **63.61**
High (2.5–4)	368	12.0		**18.52**	**2.50**; **137.45**

Bold numbers indicate significance level <0.05. ^a^
*p* value for the association between the categorized adverse work environment index and workplace bullying. ^b^ Adjusted for baseline workplace bullying and occupational level.

**Table 4 ijerph-18-10805-t004:** Associations between working conditions at baseline and workplace bullying at follow-up among employees not bullied at baseline (*N* = 1639).

	Model 1 ^a^			Model 2 ^b^		
	* **p** *	**OR**	**95% CI**	* **p** *	**OR**	**95% CI**
Organisational restructuring ^c^	0.060	1.63	0.98; 2.72	**0.037**	**1.73**	**1.03**; **2.91**
Organisational layoffs ^c^	0.438	1.23	0.73; 2.08	0.505	1.20	0.71; 2.02
Work pace ^d^	0.051	1.33	0.99; 1.76	0.069	1.30	0.98; 1.72
Amount of work ^d^	**0.004**	**1.52**	**1.15**; **2.01**	**0.000**	**1.70**	**1.27**; **2.27**
Influence at work ^d^	**0.001**	**0.62**	**0.47**; **0.83**	**0.011**	**0.68**	**0.51**; **0.92**
Role clarity ^d^	0.446	0.84	0.54; 1.31	0.466	0.85	0.54; 1.33
Quality of leadership ^d^	**0.000**	**0.58**	**0.44**; **0.77**	**0.000**	**0.58**	**0.44**; **077**

Bold numbers indicate significance level <0.05. ^a^ Each working condition unadjusted. ^b^ Additionally adjusted for occupational level. ^c^ Dichotomous measure: yes vs. no. ^d^ Scale: OR for each one-step increase in a scale score ranging from 0 to 4.

## Data Availability

A scientific use file (SUF) containing both wave 1 and wave 2 of the cohort is available at the Research Data Centre of the Federal Institute of Occupational Safety and Health.

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
