# Peer review of "Antecedents of Workplace Bullying among Employees in Germany: Five-Year Lagged Effects of Job Demands and Job Resources"

_ijerph, 2021, doi:10.3390/ijerph182010805_

Round 1

Reviewer 1 Report

The article "Antecedents of workplace bullying among employees in Germany..." is a highly interesting work regarding the causes of workplace bullying. The authors have gathered an impressive sample of employees, on which they analyse their hypothesis. From the theoretical perspective, the article is quite sound. There are however some points, which require improvement before the article can be published. I will describe them below:

  1. Authors arbitrarily decide, that workplace bullying occurs, when a person admits to an at least once-a-week occurrence of bad behaviour directed at the employee. I do not understand the basis for the decision. The reference to that point (17) refers to a mobbing paper (a separate issue) and is not elsewhere explained. This is a problem, as afterwards the authors are "forced" to use logistical regression, whereas they could transform their data about bullying to a strong, frequency-based scale (treating each occurrence of bullying as relevant and therefore not reducing the data). The authors themselves suggest that the low frequency of bullying is a problem (line 299 -> 5%), so a different (fuller) use of the available bullying data would definitely help.
  2. I miss some kind of comparison of the presented regression models. The authors build four models, with very similar construction, but fail to present the explanatory power of those model (maybe at least % of explained variance)? Without some kind of numerical indicator, it is difficult to compare the models between them. In the end, though, I would suggest using a different type of analysis, as the regression presented by the authors does not say a lot. Maybe some structural equation modelling would help?
  3. The authors write about workplace bullying, but fail to distinguish (or fail to do it clearly) between being a victim of bullying and being the perpetrator of bullying. In my opinion (which is shared in some of the papers referenced by the authors themselves) it should be distinguished as it is not the same thing (but both can be put under the term bullying).
  4. A similar issue as point 3 - the authors fail to mention the extensive literature on mobbing, a problem (or label) that may be understood as synonymous to the presented bullying. I believe that the authors should use the term they see as more appropriate in their paper, but at least some references to mobbing literature/definitions, maybe a distinction/comparison of mobbing and bullying, would help the clarity of the article.
  5. The use of JD-R as a base for analysis is good, but as such it does not give good suport to the organisational structure items used by the authors. Why should employees at different organisational levels (semi-skilled, skilled, professional... ) experience different levels of bullying? I would understand the assumption about different levels of resources, but this would/should be compensated by different levels of demands (with the end-effect same balance of the two, resulting in a similar level of expected bullying). If the authors see it differently, this is not clear from the article.

I sincerely hope that my comments will help to make the article better, as the material is definitely worth publishing. It just requires a slightly deeper /different analytical approach in order to make the conclusions (which seem plausible) stronger. 

Author Response

The article "Antecedents of workplace bullying among employees in Germany..." is a highly interesting work regarding the causes of workplace bullying. The authors have gathered an impressive sample of employees, on which they analyze their hypothesis. From the theoretical perspective, the article is quite sound. There are however some points, which require improvement before the article can be published. I will describe them below:

  1. Authors arbitrarily decide, that workplace bullying occurs, when a person admits to an at least once-a-week occurrence of bad behaviour directed at the employee. I do not understand the basis for the decision. The reference to that point (17) refers to a mobbing paper (a separate issue) and is not elsewhere explained. This is a problem, as afterwards the authors are "forced" to use logistical regression, whereas they could transform their data about bullying to a strong, frequency-based scale (treating each occurrence of bullying as relevant and therefore not reducing the data). The authors themselves suggest that the low frequency of bullying is a problem (line 299 -> 5%), so a different (fuller) use of the available bullying data would definitely help.

 Authors’ response to comment 1: Thank you for this important point. The decision to dichotomize the outcome using the daily” and “at least once a week” response options in order to identify participants exposed to bullying, is based on a widely used definition of workplace bullying, according to which the latter occurs when a person is regularly (e.g., weekly) subjected to negative acts by others at work: “Bullying at work means harassing, offending, socially exclud­ing someone or negatively affecting someone’s work. In order for the label bullying (or mobbing) to be applied to a particular activity, interaction or process it has to occur repeatedly and regularly (e.g. weekly) and over a period of time (e.g. about six months)” (Einarsen et al., 2020, p. 26; cited in the manuscript as reference number 12). We added this definition in the introduction of the revised version (see yellow-marked text from line 44, page 1 to line 52, page 2), while in the Measures section we added a sentence that explains the employed dichotomization while referring back to such definition (see yellow-marked text from line 160 to line 164, page 4). 

The use of a dichotomized version of the workplace bullying measure is not new in the epidemiological literature (e.g., Oxenstierna et al., 2012; reference number 43 added in the text and the literature list). Moreover, it has been previously shown that severe workplace bullying as represented by frequent (“daily”, “at least once a week”) exposure to bullying behaviours is associated with the most adverse effects for employees’ health status (e.g., depression; Gullander et al., 2014, reference number 4, already present in the originally submitted version). We added a sentence in the method section to clarify this point (see yellow-marked text from line 164 to line 166, page 4).

In addition, the “workplace bullying” variable is heavily skewed, given that in representative samples of the working population most of the respondents usually report a low frequency of exposure to bullying. Such distribution makes the use of workplace bullying as a frequency-based measure problematic.

We thus believe that the use of logistic regression – which has already been adopted in studies with similar designs (Oxenstierna et al., 2012; reference number 43 added in the text and the literature list) – is appropriate given the dichotomous outcomes. Given the epidemiological nature of the present study, this analytical approach provides an easy-to-interpret identification of work-related risk factors for subsequent exposure to workplace bullying. Finally, logistic regression approximates relative risks with an outcome frequency <10%, which is the case in the present study.  

The aforementioned definition of workplace bullying also highlights that the terms workplace bullying and mobbing are used interchangeably in the European tradition (see Einarsen et al. 2020), which strictly follows the early work of Leymann (1996; cited in the manuscript). See also response to point 3.

The following reference was added in the revised version:

Oxenstierna, G., Elofsson, S., Gjerde, M., Magnusson Hanson, L., & Theorell, T. Workplace bullying, working environment and health. Industrial health. 2012, 50, 180-188. https://doi.org/10.2486/indhealth.ms1300

  1. I miss some kind of comparison of the presented regression models. The authors build four models, with very similar construction, but fail to present the explanatory power of those model (maybe at least % of explained variance)? Without some kind of numerical indicator, it is difficult to compare the models between them. In the end, though, I would suggest using a different type of analysis, as the regression presented by the authors does not say a lot. Maybe some structural equation modelling would help?

Authors’ response to comment 2: Thank you for this comment. As explained in response to comment 1, we believe that a dichotomous version of the bullying measure, and the consequent use of logistic regression, are appropriate for treating the data at hand. As calculating explained variance is problematic when using logistic regression models, in the revised version we decided to calculate the attributable proportion for the main model according to the method developed by Miettinen (1974) and Rothman (2002).

The following two references were added in the revised version of the manuscript:

Miettinen, O. S.. Proportion of disease caused or prevented by a given exposure, trait or intervention. Am J Epidemiol. 1974, 99, 325-332. Reference number 49.

Rothman, K. Epidemiology: An introduction; Oxford University Press: New York, NY, USA, 2002. Reference number 50.

As a result of the added attributable proportion analysis, we included new textual parts in different sections of the revised manuscript, including abstract (see yellow-marked text in line 26, page 1), statistical analysis (see yellow-marked text from line 222, page 5 to line 236, page 6), results (see yellow-marked text from line 264 to line 275, page 6) and discussion (see yellow-marked text from line 324 to line 326, page 9). The Odds Ratios calculated as basis for the attributable proportion analysis are shown in Table 4 (page 9), which we added in the revised version of the manuscript.

To address the reviewer’s concern about the analytical models presented in Table 2, we decided to remove model 2b (model additionally adjusted for baseline workplace bullying only), since Model 3 provides the most useful information as it presents the risk of workplace bullying at follow-up after adjustment for both occupational level and workplace bullying at baseline. We believe it is worth to maintain model 2 (adjusted for occupational level), as people working in different sectors might present a different exposure to working conditions and workplace bullying, which might confound the examined relationship. Table 2 (page 8) has been revised accordingly.    

       3. The authors write about workplace bullying, but fail to distinguish (or fail         to do it clearly) between being a victim of bullying and being the                         perpetrator of bullying. In my opinion (which is shared in some of the                   papers referenced by the authors themselves) it should be distinguished as         it is not the same thing (but both can be put under the term bullying).

Authors’ response to comment 3: Thank you for this comment. In our study, we only considered workplace bullying form the perspective of the target, while it was not our intention to examine the relationship between working conditions and the risk of becoming a perpetrator. To improve clarity in the regard, we added “exposure to workplace bullying” in several parts of the text (including the abstract; see yellow-marked parts throughout the text).

           4. A similar issue as point 3 - the authors fail to mention the extensive                 literature on mobbing, a problem (or label) that may be understood as                 synonymous to the presented bullying. I believe that the authors should               use the term they see as more appropriate in their paper, but at least                   some references to mobbing literature/definitions, maybe a                                   distinction/comparison of mobbing and bullying, would help the clarity               of the article.

 Authors’ response to comment 4: Thank you for this comment. As we write in response to comment 1 above, in line with the European tradition (Einarsen et al., 2020), the terms “workplace bullying” and “mobbing” are used interchangeably, with workplace bullying having become the most common way of naming the phenomenon. Following your comment, in the revised version we added a sentence in parenthesis in the beginning of the introduction to highlight the interchangeability of the two terms (line 36, page 1). The added definition by Einarsen et al. (2020; reference 12 in the manuscript (see yellow-marked text from line 44, page 1 to line 52, page 2) also refers to such interchangeability.

           5. The use of JD-R as a base for analysis is good, but as such it does not               give good support to the organisational structure items used by the                     authors. Why should employees at different organisational levels (semi-               skilled, skilled, professional... ) experience different levels of bullying? I                 would understand the assumption about different levels of resources,                   but this would/should be compensated by different levels of demands                 (with the end-effect same balance of the two, resulting in a similar level               of expected bullying). If the authors see it differently, this is not clear                   from the article.

Authors’ response to comment 5: Thank you for your comment. In the present study, occupational level is included as covariate only (line from 203 to 207, page 5), as we were not interested in the role of occupational level as predictor of workplace bullying. Rather, in our study we aimed to examine psychosocial and organizational characteristics as potential antecedents of exposure to workplace bullying (the working conditions in focus are described in the manuscript from line 167, page 4, to line 202, page 5). To adjust for sociodemographic and occupational variables, as we did, is customary in the field as a number of these variables might be related to both the exposure and the outcome, thus potentially confounding the examined associations (see e.g. Balducci et al., 2020; Oxenstierna et al., 2012). We included occupational level as a confounder because it was significantly correlated with workplace bullying at follow-up (Appendix A) and with some of the working conditions at baseline (see Appendix B), meaning that occupational level could act as a significant confounder of the investigated association (see also response to comment 2). In the original version of the manuscript, we justified this choice with the following sentence: “We entered only occupational level and workplace bullying as baseline covariates based on the analyses shown in Appendixes A and B.” (lines 219-221, page 5).

I sincerely hope that my comments will help to make the article better, as the material is definitely worth publishing. It just requires a slightly deeper /different analytical approach in order to make the conclusions (which seem plausible) stronger. 

Authors' response: We much appreciated your comments, which we carefully considered trying to give a point-by-point answer to each of them. We hope you may find the manuscript improved. Thank you for your time reading our work.

Reviewer 2 Report

The paper deals with an important topic, one of current research interest, and it does offer new knowledge in the field of the discussed phenomena. A new element included in the paper is five-year lagged effects of job demands and job resources on workplace bullying. In general my opinion about this paper is positive.
Argumentation is solid and based on strong theoretical framework. The quantitative analysis is clear and the conclusions are substantiated by the quantitative findings. The study is located well in the literature on workplace bullying. 

The article contains numerous limitation that limit the soundness of this paper to explain the predictors and mechanisms of bullying. Despite this, it is a valuable study showing the long-term effects of job demands on organization bullying phenomena.

Overall I'm satisfied reading this paper.

Author Response

Thank you very much for your positive and appreciative comments about our manuscript.

Round 2

Reviewer 1 Report

Thanks for addressing the points. There could be still some references added, especially on using logistic regression in cases of bullying (authors suggest it's a common practice, but provide only one reference - from Oxisterna et al, where the operationalisation of bullying was different).

The remaining points are clear and do not require further changes.

Besides this issue I believe that the paper can be published.

Author Response

Dear Reviewer,

thank you for your positive feedback about the revision and for the additional comment. To address this, we replaced Oxisterna et al. (2012) with a recent systematic review of the literature about risk factors for workplace bullying (Feijó et al., 2019; reference number 17 added in the revised version of the manuscript; see yellow-marked text in lines 55, 164, 499-500). This review includes a number of studies that used a dichotomized bullying variable and a logistic regression approach to estimate the risk of bullying due to exposure to adverse working conditions. We hope this additional reference provides adequate support to our decision to dichotomize the bullying measure.  

Note that we included the new reference (Feijó et al; ref #17) in line 55 (introduction) as it further corroborates our statement the “a substantial body of research has been produced that focuses on potential work-related antecedents of workplace bullying, including job design, management practices, and the social context”. (lines 53-55).

Added reference:

Feijó, F.R., Gräf, D.D., Pearce, N., Fassa, A.G. Risk Factors for Workplace Bullying: A Systematic Review. Int J Environ Res Public Health. 2019, 16, 1945. doi: 10.3390/ijerph16111945